# Antibiotic Susceptibility and Resistance Genes in Oral Clinical Isolates of *Prevotella intermedia*, *Prevotella nigrescens,* and *Prevotella melaninogenica*

**DOI:** 10.3390/antibiotics11070888

**Published:** 2022-07-04

**Authors:** Yormaris Castillo, Nathaly Andrea Delgadillo, Yineth Neuta, Andrés Hernández, Tania Acevedo, Edwin Cárdenas, Andrea Montaño, Gloria Inés Lafaurie, Diana Marcela Castillo

**Affiliations:** 1Unidad de Investigación Básica Oral-UIBO, Vicerrectoría de Investigaciones, Facultad de Odontología, Universidad El Bosque, 110121 Bogotá, Colombia; castilloyormaris@unbosque.edu.co (Y.C.); ndelgadillos@unbosque.edu.co (N.A.D.); yneuta@unbosque.edu.co (Y.N.); lafauriegloria@unbosque.edu.co (G.I.L.); 2Facultad de Odontología, Universidad El Bosque, 110121 Bogotá, Colombia; afhernandez@unbosque.edu.co (A.H.); tacevedo@unbosque.edu.co (T.A.); edcardenasc@unbosque.edu.co (E.C.); amontanoq@unbosque.edu.co (A.M.)

**Keywords:** antibiotic susceptibility, resistance gene, *Prevotella intermedia*, *Prevotella nigrescens*, *Prevotella melaninogenica*

## Abstract

The *Prevotella* genus is a normal constituent of the oral microbiota, and is commonly isolated from mechanically treated polymicrobial infections. However, antibiotic treatment is necessary for some patients. This study compared the antibiotic susceptibility and the presence of resistance genes in clinical oral isolates of *P. intermedia*, *P. nigrescens*, and *P. melaninogenica*. Antibiotic susceptibility was assessed using the agar dilution method. PCR confirmed the species and resistance gene frequency in the *Prevotella* species. The frequencies of species *P. intermedia*, *P. nigrescens*, and *P. melaninogenica* were 30.2%, 45.7%, and 24.1%, respectively. No isolates of *P. intermedia* were resistant to amoxicillin/clavulanic acid, tetracycline, or clindamycin. *P. nigrescens* and *P. melaninogenica* were resistant to amoxicillin/clavulanic acid and tetracycline at frequencies of 40% and 20%, respectively. *P. intermedia* was resistant to metronidazole at a frequency of 30%, *P. nigrescens* at 20%, and *P. melaninogenica* at 40%. *P. nigrescens* and *P. melaninogenica* were resistant to 50% and 10% clindamycin, respectively. The gene most frequently detected was *tetQ*, at 43.3%, followed by *tetM* at 36.6%, *bla_TEM_* at 26.6%, *ermF* at 20%, *cfxA*, *cfxA*_2_, and *nimAB* at 16.6%, and *nimAEFI* at 3.3%. *P. nigrescens* was the species with the highest resistance to antibiotics such as amoxicillin/clavulanic acid, amoxicillin, and clindamycin, in addition to being the species with the largest number of genes compared to *P. intermedia* and *P. melaninogenica.*

## 1. Introduction

*Prevotella* is a genus of Gram-negative, strictly anaerobic bacteria present in the normal oral microenvironment; in polymicrobial oral infections, *Prevotella* species can function as opportunistic pathogens [1]. In the oral cavity, this genus is associated with chronic periodontitis, pulpal infections, and abscesses of dental or periodontal origin [2,3]. Such infections are treated by eradicating the etiological factors (including the infectious component) through either mechanical instrumentation, the use of chemical elements, or the use of physical agents. Thus, as an adjunctive therapy to mechanical devices, antibiotic treatment plays an essential role in the management of oral infections. The availability of such treatments has led to the development of resistance in *Prevotella* to the main antibacterial drugs used in dental practice [4].

Antibiotics from the group of beta-lactams are the first choice for treating oral infections, and tetracyclines and macrolides are the second choices. These antibiotics can eradicate most bacteria, and the incidence of adverse reactions is low. Currently, the antimicrobial susceptibility patterns of *Prevotella* species are not well understood, and existing reports have focused primarily on evaluating resistance to the beta-lactam antibiotics, which are preferred in dental practice. However, the use of combination therapy is increasing over time, leading to resistance to other families of antibiotics. The resistance of *Prevotella* to beta-lactam antibiotics was reported in 33% of cases, and resistance to clindamycin (CC) was found in 3% of cases [4]. Furthermore, the frequency of amoxicillin (AMX) resistance increased from 30% to 60% between 2011 and 2013 [5,6].

Few researchers have investigated the prevalence of antibiotic resistance genes in *Prevotella* isolated from human clinical specimens. In 2000, the presence of the *tetQ* gene was reported in the genera *Prevotella* and *Porphyromonas*; this gene confers resistance to antibiotics such as tetracyclines. In infections with anaerobic or protozoal organisms, metronidazole the treatment of choice. Its mechanism of action involves the formation of nitrous radicals that are toxic to these microorganisms. The presence of the drug may cause the expression of the *nim* gene and its allelic variants, leading to the production of enzymes that prevent the drug from forming nitrous radicals, and contributing to resistance to the drug [7,8]. A 2019 study generated great interest by reporting the prevalence of *cfxA* (50.5%), *tetQ* (30.3%), *ermF* (9.1%), and *nim* (1.0%) in *Prevotella* [9].

No studies have linked the effects of antibiotic resistance genes and susceptibility in vitro in isolates from the oral environment. Therefore, it is of great clinical importance to evaluate the presence of genes associated with antibiotic resistance and to make a comparative analysis of their effects and expression patterns by evaluating antibiotic susceptibility in the *Prevotella* genus, considering its importance in the oral cavity. In addition to the limited published scientific literature on the presence of these genetic components and their relationship with antibiotic susceptibility in these microorganisms, it is essential to have up-to-date epidemiological data on the importance of antibiotic therapy in dental practice. This study aimed to compare the antibiotic susceptibility and presence of resistance genes in *Prevotella intermedia*, *Prevotella nigrescens*, and *Prevotella melaninogenica*.

## 2. Results

### 2.1. Frequency of P. intermedia, P. nigrescens, and P. melaninogenica

Five hundred pure and viable isolates of *Prevotella* spp. were recovered; these were the general sample units. These were used to identify *P. intermedia*, *P. nigrescens*, and *P. melaninogenica*. A sample unit of 162 isolates corresponding to the identified species was used for all analyses (n = 162). Species confirmation was performed using conventional PCR. The species were identified at the following frequencies: *P. intermedia* 30.2% (n = 49), *P. nigrescens* 45.7% (n = 74), and *P. melaninogenica* 24.1% (n = 39).

Isolates were classified by periodontal clinical diagnosis; *P. nigrescens* and *P. melaninogenica* were identified in all clinical diagnosis types, including healthy, gingivitis, and periodontitis. In total, 90% of the *P. intermedia* isolates were in the periodontitis group. The species most associated with healthy states was *P. melaninogenica* 40% (Table 1).

### 2.2. Evaluation of Antibiotic Susceptibility in Isolates of P. intermedia, P. nigrescens, and P. melaninogenica

Table 2 and Figure 1 show the global antimicrobial resistance to the antibiotics AMX, amoxicillin/clavulanic acid (AMC), metronidazole (MTZ), tetracycline (TE), and CC by agar dilution method, according to the Clinical and Laboratory Standards Institute (CLSI) 2016 [10] standards for the determination of antibiotic susceptibility.

Of the isolates of *P. intermedia* (n = 10), *P. nigrescens* (n = 10), and *P. melaninogenica* (n = 10), 20%, 80%, and 30%, respectively, were resistant to AMX. No isolate of *P. intermedia* was resistant to AMC, TE, or CC. *P. nigrescens* and *P. melaninogenica* were resistant to 40% AMC and 20% TE. For MTZ, *P. intermedia* was resistant to a concentration of 30%, *P. nigrescens* to a concentration of 20%, and *P. melaninogenica* to a concentration of 40%. Finally, *P. nigrescens* was resistant to 50% CC and *P. melaninogenica* to 10% CC (Figure 1).

Table 3 shows a total of 30 *Prevotella* isolates and 10 isolates of each species. Different patterns of antibiotic resistance were also observed. For all species, 43.3% (n = 13) were resistant to β-lactam antibiotics such as AMX, and 26.6% (n = 8) were resistant to AMC. Additionally, 13.3% (n = 4) of the isolates were resistant to TE and 30% (n = 9) of the isolates were resistant to MTZ. For lincosamides, 20% (n = 6) of isolates were resistant to CC.

In the 30 *Prevotella* isolates evaluated, the inhibition range was 0.25–64 µg/mL for AMX; the minimum inhibitory concentration MIC50 was <0.5 µg/mL, while the MIC90 was 32 µg/mL. For AMC, the inhibition range was 0.25–64 µg/mL; the MIC50 was 1 µg/mL; and the MIC90 was 32 µg/mL. For TE, an inhibition range of 0.25–64 µg/mL was observed, the MIC50 was <2 µg/mL, and the MIC 90 was 16 µg/mL. For MTZ, the inhibition range was 0.25–64 µg/mL; the MIC50 was 1 µg/mL; and the MIC90 was >64 µg/mL. For CC, the inhibition range was 0.25–64 µg/mL; the MIC50 was 0.25 µg/mL; and the MIC90 > 64 µg/mL (Table 3).

### 2.3. Prevalence of Resistance Genes in P. intermedia, P. nigrescens, and P. melaninogenica

Table 4 shows the prevalence of resistance genes in the 30 isolates. Genes associated with resistance to β-lactam antibiotics (such as *cfxA*, *cfxA_2_*, and *bla_TEM_*), to tetracyclines (such as *tetM* and *tetQ*), to macrolides and CC (such as *ermF*), and to nitroimidazoles (such as *nimAB* and *nimAEFI*) were detected in isolates of *P. intermedia*, *P. nigrescens*, and *P. melaninogenica.* The most frequently detected genes were *tetQ* (in 43.3% of isolates), *tetM* (in 36.6% of isolates), *bla_TEM_* (in 26.6% of isolates), *ermF* (in 20% of isolates), *cfxA*, *cfxA_2_*, and *nimAB* (in 16.6% of isolates), and *nimAEFI* (in 3.3% of isolates).

In *P. intermedia*, the TE resistance genes *tetQ* and *tetM* were detected most frequently (40%), followed by *bla_TEM_* (20%) and *nimAB* (10%). However*, cfxA, cfxA_2_*, and *nimAEFI* were not detected. For *P. nigrescens*, *tetQ* was detected in 70% of isolates, *cfxA*, *bla_TEM_* and *ermF* in 40% of isolates, *tetM* and *nimAB* in 30% of isolates, and less frequently, *cfxA_2_* in 10% of isolates. In *P. melaninogenica*, *tetM* and *cfxA_2_* were detected most frequently (40%), followed by *bla_TEM_*, *tetQ*, and *ermF* (20%), and *cfxA* and *nimAB* (10%). *nimAEFI* was not detected.

The general frequency of resistance genes in the 162 isolates was analyzed and is included in the Appendix A.

### 2.4. Genotype–Phenotype Relationship of P. intermedia, P. nigrescens, and P. melaninogenica Isolates

The heatmap shows the genotype–phenotype relationships of the 30 isolates of *Prevotella*, and genes grouped by antibiotic class demarcated by gene names: *cfxA*, *cfxA_2_*, and *bla_TEM_* (β-lactam); *TetM* and *TetQ* (tetracyclines); *ermF* (macrolide-lincosamide); *nimAB* and *nimAEFI* (nitroimidazole). The different colors indicate: gene present and phenotype observed (red), gene present but phenotype not observed (orange), gene absent and phenotype observed (yellow), and gene absent and phenotype not observed (green) (Figure 2).

In this study, approximately 43.3% (13 out of 30) of the Prevotella isolates were simultaneously resistant to more than two antibiotics. Of the thirty isolates of *Prevotella* analyzed, a total of nine isolates did not show resistance to any of the antibiotics evaluated (30%), eight isolates were resistant to at least one antibiotic (26.6%), ten isolates showed resistance to at least two antibiotics (33.3%) and three were resistant to three antibiotics (10%).

*P. intermedia* showed a sensitive phenotype without genes associated with resistance, except for isolates 4 and 8, which presented a phenotype and genotype associated with resistance to beta-lactams and nitroimidazoles, respectively. Although isolates 9 and 10 did not present any resistance genes, they presented resistance to β-lactams and nitroimidazole.

In *P. nigrescens*, six of the ten isolates have genes for resistance to beta-lactams, tetracyclines, and lincosamides and they were alsoresistant to these antibiotics. They are indicated by a yellow profile, where resistant isolates predominate without the observed genotype. None of these isolates presented a phenotype resistant to nitroimidazoles.

Finally, in the isolates of *P. melaninogenica,* isolate 2 was the only one that did not present a resistant genotype and phenotype. Five species isolates have a beta-lactam and nitroimidazole resistant phenotype and genotype. In contrast, most of the isolates have a resistant phenotype for one of the three families of antibiotics evaluated, but do not present a genotype.

## 3. Discussion

The most frequently prescribed antibiotics in dentistry are beta-lactams, like penicillin, and amoxicillin. These antibiotics remain the most practical option to prescribe to adults for odontogenic infections. It is estimated that more than 75% of antimicrobial prescriptions are made inappropriately, and only about 50% of the patients take their medicines correctly. The widespread use of antimicrobial therapy, which is often unnecessary and inaccurate, may affect the development of microbial resistance [11].

This study investigated the extent of antibiotic susceptibility in *Prevotella* species isolated from the oral cavity, and the frequency of resistance genes present in these species. *Prevotella* species, members of the oral microbiota, are anaerobic and difficult to culture. In the 500 isolates analyzed, *P. nigrescens* was the most prevalent species, supporting previous work which isolated this species more frequently, followed by *P. intermedia* and *P. melaninogenica* [12,13,14]. However, the present study also detected other species at low frequencies, such as *P. buccae*, *P. denticola*, and *P. disiens*. These species are generally a part of the normal oral microbiota, and frequently participate in oral infections. In periodontal abscesses, 70% of the isolates corresponded to the genus *Prevotella*, and *ermF*, *cfxA*, and *nim* were frequently detected [12].

The results of the present study confirm that *Prevotella* could be an important reservoir of antibiotic resistance genes, such as *cfxA*, *cfxA*_2_, *bla_TEM_*, *tetM*, *tetQ*, *ermF*, and *nim*, which are associated with resistance to the antibiotics that are used as the first and second choices in dentistry, such as beta-lactams, macrolides, and lincosamides [4,15,16]. Additionally, the presence of these genes is related to the phenotypic results of antibiotic susceptibility.

Beta-lactamase production can be encoded by several genes, such as *cfxA/cfxA2*, *cblA*, *cepA*, *cfiA*, and *bla_TEM_* [11]. Different isolates with a resistant phenotype were presented, but they did not carry the evaluated genes. This resistance is likely mediated by other genes or allelic variants of the *cfx* gene, such as *cfxA3* or *cfxA6* [17], or other mechanisms that were not evaluated here. The antimicrobial resistance (AMR) phenotypic profiles with identified AMR genes also have been reported between the presence of *ermF*, and decreased susceptibility to azithromycin and resistance to clindamycin [18]. Isolates 3, 6, and 9 of *P. nigrescens*, presented a direct genotype–phenotype relationship due to the presence of the *ermF* gene and phenotypic resistance to clindamycin. (Figure 2).

Isolates with the presence of the evaluated genes and a sensitive phenotype were observed. These are considered essential targets, as antibiotic exposure could be a risk factor for transcribing these resistance-associated genes. These genes can be transferred to other bacteria through conjugative transposons that are activated in the presence of antibiotics. The use of antibiotics in clinical dental practice has increased, favoring the environmental pressure exerted by this increase. Antibiotic resistance among periodontal bacteria, such as *Prevotella* spp., needs to be monitored [2].

Antibiotic susceptibility was determined and correlated with resistance genes, and *P. nigrescens* was the species with the highest resistance to antibiotics such as AMX, AMC, and CC, compared to *P. intermedia* and *P. melaninogenica* [19,20]. In Colombia, Ardila et al. [21] evaluated antibiotic susceptibility to AMX, MTZ, and CC and found resistance in 35.5%, 26.6%, and 22.22% of *P. intermedia*/*nigrescens* isolates, respectively. The isolates were 100% sensitive to amoxycillin/clavulanic acid. However, antibiotic resistance in isolates of the genus *Prevotella* has increased, resulting in resistance to AMC. These results may be related to the high frequency of the antibiotic-resistant genes *cfxA*, *cfxA*_2_, and *bla_TEM_*, which have been detected in these species; these results are consistent with those reported in previous studies [22,23].

Only two isolates (heat map P.i-8 and P.m-7) of the thirty evaluated presented a phenotype and genotype resistant to metronidazole, which may be a good indicator of efficacy in current therapy. However, it is important to monitor those isolates with the genotype present and no resistance, and those that do not have the genotype present and are resistant to metronidazole, because this is the antibiotic of choice for infections involving anaerobic microorganisms.

Current evidence regarding antibiotic prescription for oral infectious processes is still scarce, limiting its management and favoring empirical prescription. Antibiotic resistance continues to be a public health problem [24]; it has increased in oral microorganisms to the extent that selective pressure is exerted at the genomic level in these species. Both in Colombia and worldwide, there are no epidemiological surveillance programs for antibiotic resistance in oral bacteria, and resistance continues to increase at an alarming rate.

This work constitutes one of the first advances in the current state of antibiotic resistance and gene frequency in species of the *Prevotella* genus in Colombia. It could serve as a basis for the future development of antibiotic management guidelines in dental practice for infections of odontogenic origin.

## 4. Materials and Methods

### 4.1. Population and Samples

In this study, we obtained 500 in vitro isolates of *Prevotella* spp; of these, 162 isolates correspond to *P. intermedia, P. nigrescens*, and *P. melaninogenica.* However, only 30 isolates were evaluated, 10 of each of the most frequent species in the oral cavity, since this genus is strictly anaerobic and, due to the rigor of the phenotypic method, it was not possible to include all of the isolates. All isolates come from the strain collection of the Oral Microbiology Laboratory of the Institute Unit of Oral Basic Investigation (UIBO). These were obtained from subgingival and other oral samples from patients who attended a consultation at the dental clinics of El Bosque University (Bogotá, Colombia).

### 4.2. Microbiological Cultures and Species Identification

Each isolate was grown in supplemented Brucella agar (0.3% Bacto agar, 0.2% yeast extract, 5% defibrinated sheep blood, 0.2% hemolyzed blood, 0.0005% hemin, and 0.00005% menadione (BBL Microbiology Systems, Cockeysville, MD, USA)) and incubated at 37 °C for 7 days under anaerobic conditions for the recovery of *Prevotella* spp. (Anaerogen, Oxoid, Hampshire, UK) [25].

### 4.3. Species Identification and Detection of Resistance Genes

*P. intermedia*, *P. nigrescens*, and *P. melaninogenica* were detected using a conventional PCR according to the recommendations of Ashimoto et al. [26]; the sequences of the primers are shown in Table 5.

The identification of each microorganism was confirmed by the presence of its amplification products on agarose gel electrophoresis at a concentration of 1.5% in Tris-acetate-EDTA (TAE) with 0.5 μg/mL of ethidium bromide. The bands were visualized using a transilluminator (Gel Doc XR+, BioRad) with ultraviolet light at 300 nm.

### 4.4. In Vitro Antibiotic Susceptibility Testing

For the determination of antibiotic susceptibility, the agar dilution method was used in this study, following the standards of the CLSI [10].

Supplemented Brucella agar (0.3% Bacto agar, 0.2% yeast extract, 5% defibrinated sheep blood, 0.2% hemolyzed blood, 0.0005% hemin, and 0.00005% menadione (BBB Microbiology Systems, Cockeysville, MD, USA)) was prepared with concentrations of antibiotics from 0.25, 0.5, 1, 2, 4, 8, 16, and 32 µg/mL up to 64 µg/mL. Antibiotics CC, TE, gentamicin, AMX, AMC, and MTZ were evaluated.

Isolates of *P. intermedia*, *P. nigrescens*, and *P. melaninogenica* were spectrophotometrically adjusted (λ = 620 nm) to a concentration of 1 × 10^8^ bacteria/mL (optical density). These inocula were used to determine the MIC of each antibiotic evaluated in this study. Subsequently, the culture plates inoculated with each isolate in each of the different antibiotic concentrations were incubated at 37 °C for 7 days in anaerobiosis (Thermo Scientific: AnaeroGen Oxiod Ltd., Basingstoke, UK; Anaerobic indicator BR0055B, Oxiod Ltd., Basingstoke, UK). After incubation, MIC values for each antibiotic were evaluated in the *P. intermedia*, *P. nigrescens*, and *P. melaninogenica* isolates, with their respective dilutions. The first agar box with an antibiotic concentration at which no bacterial growth was observed was determined as the MIC. The reference strains *B. fragilis* ATCC 25285, *P. intermedia* ATCC 25611, *P. nigrescens* ATCC 33561, and *P. melaninogenica* ATCC 25845 were used as the internal controls.

### 4.5. Gene Detection

The presence of *cfxA* and *cfxA_2_* was determined by conventional PCR, as previously described by Fosse et al. [27]. The *bla_TEM_* gene was detected according to the protocol specified by Ioannidis et al. [28]. To determine the presence of *tetM* and *tetQ* genes, the protocol specified by Lacroix et al. was used with some modifications [29,30]; the *nimAB*/*AEFI* gene was also identified using this method. The reaction occurred in a final volume of 25 µL, of which 5 µL corresponded to the sample and 20 µL to the reaction mixture containing 1X PCR buffer (1.0 mM KCl, 10 mM Tris-HCl (pH 9, 0 to 25 °C), 2.0 mM MgCl_2_, and 0.1% Triton^®^ X-100); 0.04 IU Taq DNA polymerase (Promega, Madison, WI, USA); 1.5 mM MgCl_2_; 0.02 mM of each deoxyribonucleotide and 2 µM of each primer. To determine the presence of the *ermF* gene, the protocol described by Reig et al. [31] was implemented.

### 4.6. Statistical Analysis

Databases were built in Microsoft Office Professional Plus Excel 2013 and were used to calculate the frequencies of *P. intermedia*, *P. nigrescens*, and *P. melaninogenica*, the frequency of resistance genes, and antibiotic resistance (MIC 50 and MIC 90) to perform a descriptive analysis and express results in percentage data.

## 5. Conclusions

*P. nigrescens* was the species with the highest level of resistance to antibiotics such as AMC, AMX, and CC; it also possessed resistance genes at a higher frequency compared to *P. intermedia* and *P. melaninogenica*. AMX was the antibiotic to which the species were least susceptible; it was followed by AMC, CC, and MTZ. This information may be relevant for the management and treatment of oral infections caused by *Prevotella*.

## Figures and Tables

**Figure 1 antibiotics-11-00888-f001:**
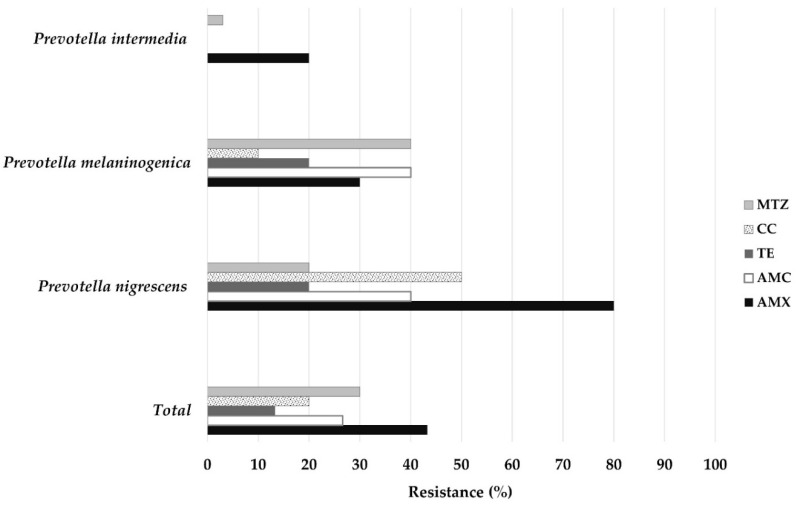
Frequency of resistance of *Prevotella nigrescens, P. intermedia*, and *P. melaninogenica* to the antibiotics evaluated.

**Figure 2 antibiotics-11-00888-f002:**
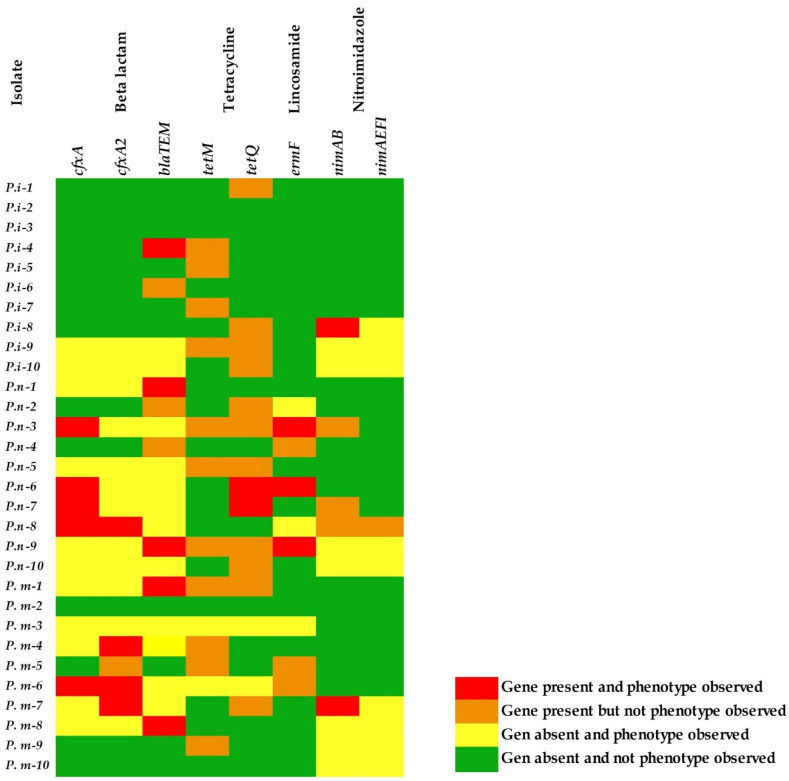
Heat map of the summary of the genotype–phenotype relationships of the *P. intermedia*, *P. nigrescens*, and *P. melaninogenica* isolates.

**Table 1 antibiotics-11-00888-t001:** Distribution of isolates by species and periodontal diagnosis.

	Healthy (%)	Gingivitis (%)	Periodontitis (%)
*P. melaninogenica*	40	30	30
*P. nigrescens*	10	40	50
*P. intermedia*	0	10	90

**Table 2 antibiotics-11-00888-t002:** Global resistance and minimum inhibitory concentration (MIC) ranges, MIC50, and MIC90 against *Prevotella nigrescens, P. intermedia*, and *P. melaninogenica*.

Global Resistance					
Antibiotics	n	Range ^1^ (µg/mL)	MIC 50 ^2^ (µg/mL)	MIC 90 ^3^ (µg/mL)	R
					n (%)
AMX	30	<0.25–>64	0.5	32	13 (43.3)
AMC	30	<0.25–>64	1	16	8 (26.6)
TE	30	<0.25–>64	2	16	4 (13.3)
CC	30	<0.25–>64	0.25	>64	6 (20)
MTZ	30	<0.25–>64	1	>64	9 (30)

AMX: amoxicillin; AMC: amoxicillin/clavulanic acid; TE: tetracycline; CC: clindamycin; MTZ: metronidazole. ^1.^ Antimicrobial range in which inhibition was detected (µg/mL); ^2.^ Minimum Inhibitory Concentration for 50% of isolates; ^3.^ Minimum Inhibitory Concentration for 90% of isolates; S/R sensitivity and resistance range (µg/mL) according to the CLSI (Clinical & Laboratory Standards Institute, 2016) [10] for each antibiotic evaluated.

**Table 3 antibiotics-11-00888-t003:** Minimum inhibitory concentration (MIC) ranges and MIC50 and MIC90 for the tested antimicrobial drugs against *Prevotella nigrescens, P. intermedia*, and *P. melaninogenica* in oral isolates.

** *Prevotella intermedia* **					
**Antibiotics**	**n**	**Range ^1^ (µg/mL)**	**MIC 50 ^2^ (µg/mL)**	**MIC 90 ^3^ (µg/mL)**	**R**
					**n (%)**
AMX	10	<0.25–8	<0.25	4	2 (20)
AMC	10	<0.25–2	<0.25	1	0
TE	10	<0.25–4	<0.25	4	0
CC	10	<0.25–4	<0.25	1	0
MTZ	10	<0.25–>64	1	>64	3 (30)
** *Prevotella nigrescens* **					
**Antibiotics**	**n**	**Range ^1^ (µg/mL)**	**MIC 50 ^2^ (µg/mL)**	**MIC 90 ^3^ (µg/mL)**	**R**
					**n (%)**
AMX	10	<0.25–>64	16	64	8 (80)
AMC	10	0.5–>64	4	64	4 (40)
TE	10	<0.25–16	2	16	2 (20)
CC	10	<0.25–>64	1	>64	5 (50)
MTZ	10	<0.25–64	1	32	2 (20)
** *Prevotella melaninogenica* **					
**Antibiotics**	**n**	**Range ^1^ (µg/mL)**	**MIC 50 ^2^ (µg/mL)**	**MIC 90 ^3^ (µg/mL)**	**R**
					**n (%)**
AMX	10	<0.25–32	0.25	16	3 (30)
AMC	10	<0.25–64	2	32	4 (40)
TE	10	<0.5–>64	2	64	2 (20)
CC	10	<0.25-64	0.25	1	1 (10)
MTZ	10	<0.25–>64	2	>64	4 (40)

AMX: amoxicillin; AMC: amoxicillin/clavulanic acid; TE: tetracycline; CC: clindamycin; MTZ: metronidazole. ^1^ Antimicrobial range in which inhibition was detected (µg/mL); ^2^ Minimum Inhibitory Concentration for 50% of isolates; ^3^ Minimum Inhibitory Concentration for 90% of isolates; S/R sensitivity and resistance range in (µg/mL) according to the CLSI (Clinical & Laboratory Standards Institute, 2016) [10] for each antibiotic evaluated.

**Table 4 antibiotics-11-00888-t004:** Frequency of genes *cfxA, cfxA_2_, bla_TEM_, tetM, tetQ, ermF, nimAB,* and *nimAEFI* in 30 oral isolates of *Prevotella intermedia, P. nigrescens,* and *P. melaninogenica*.

Bacteria	n	*cfxA* F (%)	*cfxA_2_* F (%)	*blaTEM* F (%)	*tetM* F (%)	*tetQ* F (%)	*nimAB* F (%)	*nimAEFI* F (%)	*ermF* (%)
*P. intermedia*	10	0	0	2 (20)	4 (40)	4 (40)	1 (10)	0	0
*P. nigrescens*	10	4 (40)	1 (10)	4 (40)	3 (30)	7 (70)	3 (30)	1 (10)	4 (40)
*P. melaninogenica*	10	1 (10)	4 (40)	2 (20)	4 (40)	2 (20)	1 (10)	0	2 (20)
Total	30	5 (16, 6)	5 (16, 6)	8 (26, 6)	11 (36, 6)	13 (43, 3)	5 (16, 6)	1 (3, 3)	6 (20)

**Table 5 antibiotics-11-00888-t005:** Primer sequences for detecting species and resistance genes.

Gen	Sequence (5′→3′)	Amplicon Size (bp)	Reference
16S—*P. intermedia **	TTTGTTGGGGAGTAAAGCGGG TCAACATCTCTGTATCCTGCGT	575	[26]
16S—*P. nigrescens **	ATGAAACAAAGGTTTTCCGGTAAG CCCACGTCTCTGTGGGCTGCGA	804	[26]
16S—*P. melaninogenica **	TACAATGGAGAGTTTGATCC CGATCCTTGCGGTCACGGAC	1453	This study
*cfxA* **	GCAAGTGCAGTTTAAGATT GCTTTAGTTTGCATTTTCATC	934	[27]
*cfxA_2_* **	CAAAGYGACAAYAATGCCTGCG TSACGAAGRCGGCWAT	426	[27]
Bla_TEM_ ^†^	ATGAGTATTCAACATTTCCG CCAATGCTTAATCAGTGAGG	858	[28]
*tetM* ^††^	GACACGCCAGGACATATGG TGCTTTCCTCTTGTTCGAG	397	[29]
*tetQ ***	GGCTTCTACGACATCTATTA CATCAACATTTATCTCTCTG	755	[30]
*ermF*	TTTCGGGTCAGCACTTTACTA ACTTTCAGGACCTACCTCATA	476	[31]
*nimAB* ^††^	GGCTACAAGCAGCATGTC TGCATACTTTGCTCTTC	377	This study
*nimAEFI* ^††^	TGCATACTTTGCTCTTC ATGTTCAGAGAAATGCGGCG	455	This study

* Species positive control: DNA of *P. intermedia* ATCC 25611, *P. nigrescens* ATCC 33561, and *P. melaninogenica* ATCC 25845. Positive gene controls: ** *Bacteroides fragilis* ATCC 25285 (*cfxA, cfxA_2_, tetQ*), ^†^ *Escherichia coli* ATCC 25922 (*bla_TEM_*), ^††^
*Fusobacterium nucleatum* ATCC 25586 *tetM*, *nimAB*/*AEFI*; negative control: molecular grade water.

## Data Availability

Not applicable.

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
