# Peer review of "Antibiotic Susceptibility and Resistance Genes in Oral Clinical Isolates of Prevotella intermedia, Prevotella nigrescens, and Prevotella melaninogenica"

_antibiotics, 2022, doi:10.3390/antibiotics11070888_

Round 1

Reviewer 1 Report

Prevotella is a genus of gram-negative anaerobic bacteria in normal oral microbiota. Authors compared the antibiotic susceptibility and the presence of resistance genes in clinical oral isolates of P. intermedia, P.nigrescens, and P.melaninogenica. The results indicated P. nigrescens was the species with the highest resistance to antibiotics as amoxicillin/clavulanic, amoxicillin, and clindamycin, and the presence of resistance genes compared to P. intermedia and P.melaninogenica. the results are interesting and important.

There are some comments as below:

1. L36-38, which widespread use please provide.

2. L39: Beta-lactams, tetracyclines, and macrolides are the first and second choices, please check.

3. the results part is better.

4. the materials and methods part is better.

5. the discussion part is better, if authors compared with this genus, maybe will provide a guide to oral.

Author Response

Dear reviewer.

Below we send the answers to your comments. Additionally, the manuscript is attached with the changes in red, in these are the changes suggested by the three evaluators, so you will find some additional changes to those requested by you.

Thank you very much for your contributions, which improve our work.

Cordially,

Diana Marcela Castillo

1. L36-38, which widespread use please provide.

Answer: In the L-36-L38 we change the word widespread by provide

2. L39: Beta-lactams, tetracyclines, and macrolides are the first and second choices, please check.

Answer: the wording was adjusted in the manuscript.

Reviewer 2 Report

This manuscript by Yormaris Castillo et al. describes the study of antibiotic resistance in Prevotella spp. by genetic and phenotypic analyses.

This manuscript describes very partial results and the study needs to be completed before possible resubmission.

The major point is that the authors have studied only 30 strains and not their entire biocollection. without any justification (economic?) The authors can do antibiotgrams and targeted genetic research on the rest to have a more complete view. Moreover, and this is the interest of an academic center, it is necessary to retrieve the clinical information of these patients, to anticipate a possible exposure to antibiotics in the life of the patients.

Other comments below: 

Global: italicize "in vitro", "et al".

Tables 1 and 2 would benefit from being represented as a histogram.

Table 3: the authors should study the genotype-phenotype relationship.

Methods: did the authors ensure that the antibiotic did not degrade in the agar? 

Author Response

Dear reviewer.

Below we send the answers to your comments. Additionally, the manuscript is attached with the changes in red, in these are the changes suggested by the three evaluators, so you will find some additional changes to those requested by you.

Thank you very much for your contributions, which improve our work.

Cordially,

Diana Marcela Castillo

This manuscript describes very partial results, and the study needs to be completed before possible resubmission.

The major point is that the authors have studied only 30 strains and not their entire biocollection. without any justification (economic?) The authors can do antibiograms and targeted genetic research on the rest to have a more complete view.

Answer: The results of this work do correspond to a study of a biocollection. However, even though it started from 162 isolates, only 30 isolates were evaluated for the agar diffusion method to determine antibiotic resistance and detect resistance genes. Due to the difficulty in growing these strictly anaerobic microorganisms, the rigor of the agar dilution method for anaerobic microorganisms was not possible to evaluate more isolates. Ten isolates of each of the most frequent species in the oral cavity can be evaluated for all antibiotics and resistance genes. Table 1S (supplementary material) reports the presence of the genes evaluated in the 162 isolates.

Moreover, and this is the interest of an academic center, it is necessary to retrieve the clinical information of these patients, to anticipate a possible exposure to antibiotics in the life of the patients.

Answer: Unfortunately, we do not have patient information to confirm patients' prior antibiotic exposure. However, we collected the periodontal diagnostic data that we will include in the manuscript. We can confirm that since they are mostly isolated from healthy individuals or patients with periodontitis, they were not under antibiotic treatment at the time the sample was likely taken. However, we do not have the information to confirm it.

A table of the distribution of isolates by species and periodontal diagnosis was included (table 1).

Other comments below: 

Global: italicize "in vitro", "et al".

Answer: It was adjusted in the manuscript

Tables 1 and 2 would benefit from being represented as a histogram.

Answer: A figure with the frequency of resistance for each species studied is included (Figure 1). However, we consider tables 1 and 2 crucial because of the MIC information they present and because the other evaluators agree with the information offered.

Table 3: the authors should study the genotype-phenotype relationship.

Answer: We find your observation very pertinent; for this reason, we include a heat map with these data in the manuscript, and we expand this analysis a little more both in the item of results and in the discussion (Figure 2).

Methods: did the authors ensure that the antibiotic did not degrade in the agar? 

Answer:  To validate the method and confirm the antibiotic did not degrade, the reference strain Bacteroides fragilis ATCC 25285 was used as MIC quality control. The acceptable range for each antimicrobial was validated according to by the CLSI 2016.  AMX 16-64µg/mL AMC 0.25-1 µg/mL, CC 0.5-2µg/mL, MET 0.25-1 µg/mL, TE 0.12-0.5µg/m. In addition, the temperature of the medium was controlled to avoid the degradation of the antibiotics.

Reviewer 3 Report

This article by Yormaris Castillo and colleagues summarizes the antibiotic susceptibility and prevalence of resistance genes in different species of Prevotella. The manuscript is very well-organized and results are clear. Nevertheless,  the paragraphs are quite short and miss transitions to guide the reader. In addition, it would be appreciated to include a brief note/decription about the method, to avoid going back and forth with the Materials and methods section (such as species typing by 16S amplification etc..).

In the end, a summary with percentages, like a graph, would be useful. It is only described in the text and is difficult to put in perspective and compare phenotypes with identified genes because only mentioned in the text, and briefly in the discussion.

One of the point being the phenotypic observation for antibiotic resistance and resistance gene content, a dedicated paragraph and figure is necessary. I think this analysis needs to be performed in depth. The authors only considered the populations for the different species, do not mention any genotyping approach. The authors don't mention if the isolates showing resistance do have or not any typable resistance gene. Is there any discordance or limitations, considering the resistance genes known to date?

Author Response

Dear reviewer.

Below we send the answers to your comments. Additionally, the manuscript is attached with the changes in red, in these are the changes suggested by the three evaluators, so you will find some additional changes to those requested by you.

Thank you very much for your contributions, which improve our work.

Cordially,

Diana Marcela Castillo

This article by Yormaris Castillo and colleagues summarizes the antibiotic susceptibility and prevalence of resistance genes in different species of Prevotella. The manuscript is very well-organized and the results are clear. Nevertheless, the paragraphs are quite short and miss transitions to guide the reader. In addition, it would be appreciated to include a brief note/description about the method, to avoid going back and forth with the Materials and methods section (such as species typing by 16S amplification, etc).

In the end, a summary with percentages, like a graph, would be useful. It is only described in the text and is difficult to put in perspective and compare phenotypes with identified genes because only mentioned in the text, and briefly in the discussion.

One of the point being the phenotypic observation for antibiotic resistance and resistance gene content, a dedicated paragraph and figure is necessary. I think this analysis needs to be performed in depth. The authors only considered the populations for the different species, do not mention any genotyping approach. The authors don't mention if the isolates showing resistance do have or not any typable resistance gene. Is there any discordance or limitations, considering the resistance genes known to date?

Answer:

A more detailed description of the results is made, to guide the reader in the main findings.

Following the guidelines of the journal, the methods are included at the end of the manuscript, however, to make reading practical, a brief methodological summary is included for each result.

We find your observation very pertinent. For this reason, we include in the manuscript a figure with the frequency of resistance for each of the species studied is included (Figure 1), and a heat map genotype-phenotype relationship evaluated (Figure 2), with these data we expand this analysis a little more both in the item of results and in the discussion.

Round 2

Reviewer 1 Report

The version is better, I recommend accept in present form.

Reviewer 2 Report

The manuscript has been revised according to my previous comments.